



# Determining the susceptibility of soils materials to erosion by rain-
# impacted flows
P.I.A. Kinnell
Institute for Applied Ecology, University of Canberra, Australia
Correspondence to: P.I.A. Kinnell ( peter.kinnell@canberra.edu.au)
## Abstract
Conceptually, rain has a capacity to cause erosion (rainfall erosivity) and soils have a susceptibility to
erosion by rainfall (soil erodibility) but no absolute measure of rainfall erosivity exists. Consequently,
soil erodibility is nothing more than an empirical coefficient in the relationship between an index of
rainfall erosivity and soil loss. Erosion by rain-impacted flow is influenced by the size, velocity and
impact frequency of the raindrops but also flow depth and velocity. Experiments with artificial rainfall
falling on sloping surfaces in the field usually do not enable flow depth and velocity to be well
measured or controlled. Also, sprays produce artificial rainfall where the spatial uniformity in rainfall
intensity, drop size and frequency is often less than desirable. Artificial rainfall produced by pendant
drop formers can produce rainfall that has better spatial uniformity. Equipment for controlling flow
depth and velocity over eroding surfaces has been developed and used to calibrate the effect of flow
depth on the discharge of sediment by rain-impacted flow using artificial rainfall having a uniform
drop-size distribution under laboratory conditions. Once calibrated, laboratory experiments can be
conducted to rank soils according to their susceptibility to erosion under the flows impacted by the
artificial rainfall under conditions where the erosive stress applied to the eroding surface is well
controlled.
Key Words: Rainfall erosion; soil erodibility; laboratory experiments; rainfall uniformity.



**1 Introduction**

Conceptually, rain has a capacity to cause erosion (rainfall erosivity) and soils have a
susceptibility to erosion by rainfall (soil erodibility). Rainfall erosion is a complex process and, in
reality, no absolute measure of rainfall erosivity exists. Various parameters (eg. rainfall kinetic energy)
and combinations of parameters (eg. product of storm rainfall energy and the maximum 30-minute
intensity) are used as indices of rainfall erosivity and, as a consequence, soil erodibility is nothing more
than an empirical coefficient in the relationship between an index of rainfall erosivity and soil loss. Also
the ranking of soils according to their erodibility values can vary depending on how those erodibilities
are determined experimentally and the model being used. For example, Kinnell and Risse (1998)
presented soil erodibility values in SI units obtained using runoff and soil loss plot data at a number of
locations in the USA for the Universal Soil Loss Equation (USLE; Wischmeier and Smith, 1965, 1978),
which uses the $EI_{30}$ as the event erosivity index, and the USLE-M which uses the product of the runoff
ratio ($Q_R$) and $EI_{30}$. Table 1 shows the ranking obtained using the soil erodibilities for the USLE (K)
published by Risse et al (1993) and the erodibilities observed Kinnell and Risse for the USLE and the
USLE-M ($K_{UM}$) for the bare fallow plots at 14 locations. All three rankings are different. In addition
field experiments designed to determine K factor values using artificial rainfall in the USA measure soil
loss under different moisture states (dry, wet, very wet) and weight the result so as to provide an
estimate of K for central USA (Romkens, 1985). Consequently, K factor values determined using the
soil erodibility nomograph (Wischmeier et al, 1971) need to be adjusted for the difference in climate
when applied outside that geographic area (USDA, 2008).

Modern understanding of rainfall erosion recognizes that rainfall erosion is caused by the
expenditure of the kinetic energies of raindrops and surface water flows acting either singly or together.
Detachment of soil material from the soil surface is essential before the transport of soil material away
from the site of detachment leads to soil erosion. Detachment in sheet and interrill erosion which erode
the most chemically active layer of the soil, the soil surface, is dominated by the expenditure of the




kinetic energy of the impacting raindrops. Raindrop impact is also involved in transporting soil particles
in rain-impacted flows. The WEPP model ((Flanagan and Nearing, 1995) was developed to predict soil
loss by modelling flow driven erosion in rills and raindrop driven erosion in interrill areas as separate
processes. In the WEPP  interrill erosion model, the event erosivity index is the product of the runoff
rate ($q_w$) and rainfall intensity (I) so that $D_s$ (mass/area/time), the rate sediment is discharged from the
interrill area, is given by
$\qquad D_s = k_i\ q_w\ I\ S_f$ $\hfill$ (1)
where, $k_i$ is the interrill erodibility and $S_f$ is a function of slope gradient. However, the ranking of the
soils in the field experiments of Elliot (1989) depends on whether the plots had inclined surfaces similar
to those used in ridge tillage agricultural systems or were relatively flat (Table 2). The correlation
between the interrill erodibilities on the flat and ridged plots was also poor (Figure 1). One reason for
this is that sediment discharge rate ($q_s$, , mass/width/time) generated by raindrop-induced saltation that
often controls erosion on sheet and interrill areas is influenced by a water depth (h), drop size and flow
velocity (u),
$\qquad q_s(h,d) = k_{sd}\ u\ I_d\ f[h,d]$ $\hfill$ (2)
where $k_{sd}$ is a coefficient that depends the characteristics of material being eroded, u is flow velocity, $I_d$
is the intensity of the rain made up of drops of size d, and f[h,d] is a function  accounting for the effect
on sediment discharge of the interaction between flow depth (h) and drop size (Kinnell, 1993b), but the
effects of  u, $I_d$, h and d are not taken into account in Eq. (1). Many subsequent experiments have been
performed in many parts of the world, in some cases, the surfaces have been inclined at angles
commonly observed in ridge tillage systems used in agriculture, but in many cases, relatively flat
surfaces have been used.  Consequently, given the results presented in Table 2, misleading modelled
results may have been produced using interrill erodibilities from many experiments because of the lack
of adequate control of the factors that influence soil loss generated by rain-impacted flows in those
experiments.

$\qquad$ Rain-impacted flows dominate erosion in sheet and interrill erosion areas. As noted above,
these areas are important given that erosion removes soil from the soil surface and usually, the soil



surface is the most chemically rich and active part of the soil. Consequently, it is important to determine
the susceptibility of soil surfaces to erosion by rain-impacted flows  when the factors that influence soil
loss generated by rain-impacted flows are known and well controlled. As demonstrated here, this is
possible under laboratory conditions.

**2 Equipment**

The objective of the equipment described here is to produce the situation where an

erodible surface is eroded by rain-impacted flow where flow depth and velocity are controlled in order
to control the erosive stress applied by the impacting raindrops to the eroding surface. The apparatus
shown in Figure 2 is a modification of the one originally developed by Moss and Green (1983).
Basically, the erodible material is contained in a box within a flume with an adjustable weir to facilitate
the control of flow depth and velocity. A ripple guard is placed at the downstream end to prevent ripples
produced by drop impacts affecting the capacity of the weir to control flow depth and velocity. Inflows
are controlled so that, together with adjustments to the height of the weir, flow depths and velocities can
be set. A rain bleed is used to bleed water from the flume when rainfall is applied in order to maintain
the outflow at the inflow rate. This bleed is of major importance in obtaining flow depth and velocity
control when flows are very shallow. In experiments using sand by Moss and Green (1983) and Kinnell
(1988, 1993b), the eroding area was 500 mm by 500 mm. In the experiments by Kinnell, flow depth
prior to rain was measured by having a removable horizontal bridge above the bed and measuring the
distance from the top of the bridge to the eroding surface when there was no flow, and the distance to
the water surface when the was flow using a vernier depth gauge with a sharp pointed end. Contact with
the water and erodible surfaces was detected electrically. A pressure port located at the bottom end of
the sand box enabled the flow depth to be monitored during rain when eroding sand (Kinnell, 1988).

Often, laboratory experiments on rainfall erosion involve packing loose soil material into boxes

and exposing them to rain. Loose soil can packed into the box as an alternative to sand in order to erode



soil under controlled flow conditions. However, a version of the apparatus has also been developed that
enables experiments to be undertaken using intact soil monoliths having a surface area 250 mm wide by
500 mm long (Kinnell and McLachlan, 1989). Experiments have been performed on surface soils from
Ginninderra near Canberra, Australia and near Coolabah in New South Wales using this apparatus.
Metal sampling frames as depicted in Figure 3 were used to collect blocks of soil 500 mm long, 250
mm wide and 100 mm deep. The area to be taken was dug around to produce a block of soil a little
bigger than 500 mm by 250 mm by 100 mm deep. The frame was bevelled on the bottom edged to aid
cutting the soil as it was pushed down over the block. Once in place, 125 mm by 5.6 mm nails were
then hammered through the sides to provide a support structure. The bottom of the block was then cut to
free the monolith and the frame tipped gently on its side. The bottom of the monolith was then trimmed
square with the base and the monolith tipped back onto a supporting base of marine ply having an area a
little larger than 500 mm by 250 mm. A similar piece of wood was placed on the top of the frame and
straps used to make a package for transport. Prior to the experiments with rain-impacted flow, the top
piece of wood was removed and the frame containing the soil supported by the marine ply base placed
in the flume. The monolith was then saturated by wetting from the bottom. Each monolith was
subjected to 10 minute of rainfall with d = 2.7 mm produced 11.2 m above the surface and a rainfall
intensity of 64 mm/h with flow depth at the maximum and u = 20 mm/s as a pre-treatment. Then 10
mins of rain was applied at each of 4 flow depths with $I_d$ = 64 mm/h and u = 20 mm/s with flow depths
decreasing over the range 7.5 mm to 3.2 mm.

Some soil particles detached by raindrops impacted flows travel in the flow in suspension while

others saltate or roll along the surface. Saltation and rolling can occur in unimpacted flows only if they
have flow velocities that exceed certain critical values. Below these critical flow velocities, soil
particles may be induced saltate or roll only in association with the impact of individual raindrop
impacts. Flow velocities below those causing flow driven saltation and rolling are common in many
sheet and interrill erosion areas and consequently, the experiments were undertaken at flow velocities
set about 20 mm/s so that only raindrop driven saltation and rolling could occur for particles not
travelling in complete suspension.




Modules producing pendant raindrops from hypodermic needles 11.2 m above the eroding
surface produced the artificial rainfall applied in the experiments reported by Moss and Green (1983)
and Kinnell (1988, 1993b).  The modules produced drops over an area 1 m by 1 m. The needles were
spaced 25 mm apart. Rainfall intensity was controlled using a metering pump because previous work
established that using hydrostatic pressure as a mechanism to control rainfall intensity did not provide
good temporal control of rainfall intensity. The modules were moved back and forth 250 mm along the
line of flow at 4.2 mm/s. Spatial variations in rainfall intensity do occur in rain produced from
hypodermic needles due to variations in the drop production rate between nearby needles and variations
in the trajectory of drop fall. Moving the modules back and forth helps produce high levels of spatial
uniformity in rainfall intensity which are important in facilitating the movement of particles travelling
by raindrop induced saltation and rolling (Kinnell, 1991). In raindrop induced saltation and rolling
particles move a limited distance from the point of drop impact and remain sitting on top of the soil
matrix until disturbed again by a subsequent drop impact. Consequently, any positive or negative
variation from the mean in respect to drop size, drop velocity and drop impact frequency at the
downstream end of an eroding surface will result in sediment discharges that differ from those where
these factors are completely uniform spatially.

**3 Calibration**

In order to determine soil erodibilities using the apparatus described above, equations that
describe the flow depth – drop size function in Eq. (1) specifically for the experimental conditions are
needed particularly when, as is often the case in many laboratory situations, drops are travelling at
considerably less than terminal velocity. The procedure for setting flow depth involves setting a height
for the weir and running a set of inflow rates and measuring the flow depths associated with them.
Given data on flow depth and inflow rate, flow velocity is then calculated and the inflow rate meeting
the desired flow criteria selected for the erosion experiments.





Prior to undertaking the experiments to calibrate flow depth and velocity, the erodible surface

should have been prepared and put in place. Experiments for determining f[h,d] for drops travelling at
or near terminal velocity by Kinnell (1991, 1993b) were undertaken using beds of 0.2 mm sand, leveled
and smoothed prior to flow conditions being set. 0.2 mm sand was also used for the experiments at
subterminal velocity (Kinnell, 2005a). Beds of uniform sized sand are best to use since all particles have
the same potential rate of travel. The duration of exposure to rain was arbitrarily set at 10 minutes and
gave measureable quantities of eroded material without a loss of material sufficient to change the height
of the surface appreciably from the original height over the bottom half of the eroding surface. Although
0.2 mm was used here in the calibration exercise, Eq. 2 has been shown to apply to the erosion of beds
of sand containing particles ranging in size between 0.1 mm and 0.9 mm (Kinnell 1991, 1993b).

The experiments undertaken by Kinnell (1991, 1993b) showed that the rate sand was discharged

varied linearly with rainfall intensity and flow velocity as indicated in Eq.(2). Sediment discharge ($q_s$,
mass/width/time) is given by the product of water discharge ($q_w$, volume/width/time) and sediment
concentration ($c_s$), the mass of material discharged with the flow per unit volume of water;

$q_s = q_w\, c_s$        (3)


so that

$c_s = q_s\, /\, q_w$        (4)


and, from Eq. (1),

$c_{sd} = k_{sd}\, I_d\, f[h,d]/h$        (5).


where $c_{sd}$ is the sediment concentration produced by the impact of drops of size d and $k_{sd}$ is a coefficient
related to the susceptibility of the eroding material to erosion. Dividing both sides of Eq. (5) by $I_d$ gives






$$c_{sd}/I_d = k_{sd} \, f[h,d]/h \tag{6},$$

Consequently, for an eroding surface of uniform sized sand, using sediment concentration per unit
rainfall intensity as the dependent variable provides an equation where the effect of the drop size - flow
depth interaction can be evaluated even when rainfall intensity and flow velocity vary. Figure 4 shows
the relationship between sediment concentration per unit rainfall intensity and flow depth obtained for
2.7 mm drop impacting flows at close to terminal velocity over beds of 0.2 mm sand in experiments
reported by Kinnell (1991, 1993b). As illustrated by Figure 5, the relationship between sediment
concentration per unit rainfall intensity and flow depth determined for 0.2 mm sand provides a
mathematical expression that is linearly related to sediment concentration per unit rainfall intensity for
sediment discharged from other sand surfaces and this leads to the equation

$$c_{sd}/I_d = k_s \, (0.0015 \, h^2 - 0.0291 \, h + 0.1443) \quad , d=2.7, h<9 \text{ mm} \tag{7}$$

where $k_s$ varies with the particle size. In addition, Kinnell and McLachlan (1989) showed that the form
of the relationship between sediment concentration and flow depth when rain-impacted flows erode
cohesive soil surfaces was the same as that for sand. However, as indicated by Figure 6, a different
calibration equation is required when drop size and velocity conditions vary from those used in the
experiments that produced Eq. (7).






**4 Ranking surface susceptibility to erosion by rain-impacted flow.**

In Eq. (7), $k_s$ for 0.2 mm sand is, in theory, equal to 1.0 so that the $k_s$ values for other sized sands
are scaled relative to the susceptibility of 0.2 mm sand to erosion by flows impacted by 2.7 mm drops
travelling at near terminal velocity. A more general equation for the effect of drop size and eroding
material on the ratio of $c_{sd}$ to $I_d$ is given by

$$c_{sd}/I_d = k_{sd} (a_d\, h^2 - b_d\, h + 1.0) \qquad\qquad (8)$$

where $k_{sd}$ acts as an index of the susceptibility of the eroding surface to erosion by drops of size d, and
$a_d$ and $b_d$ are coefficients that vary with d. For 2.7 mm drops travelling at near terminal velocity and h<
9mm, $a_d = 0.0104$ and $b_d = 2.017$. Figure 7 shows the application of Eq. (8) when surfaces of soil
monoliths were exposed flows impacted by 2.7 mm raindrops travelling at near terminal velocity in the
modified version of the apparatus shown in Figure 1. As noted above, the Ginninderra soil monolith
came from bare fallow runoff and soil loss plots near Canberra, Australian Capital Territory (Kinnell,
1983) while the Oakvale monoliths came from a location near Coolabah, New South Wales, Australia
and had differing levels of cryptogamic crust cover. $k_{sd}$ in Eq. (8) is an index of the susceptibility of the
soil to erosion only in a qualitative sense.

In the experiments preformed with sand, new surfaces were prepared for each rainfall event. In
the experiments with soil monoliths, each monolith was eroded by 4 rainfall events starting with 2
rainfall events at the maximum flow depth used. The first event was a pre-treatment but in reality, each
rainfall event that precedes another is a pre-treatment to subsequent events. Starting the sequence from
the shallowest flow produces a different result (Kinnell et al., 1996).

Over a limited range of flow depths, a linear relationship also exists with the inverse of flow
depth minus 0.1 when the 2.7 mm drops fall from 11.2 m (Figure 8);




$$c_{sd}/I_d = k_{1sd} / (1/h - 0.1) \qquad , d=2.7, \ 3d < h > 4mm \qquad (9)$$

where $k_{1sd}$ is a coefficient that is related to the susceptibility of the soil to erosion by rain impacted flow.
The upper depth limit is, in the case of drops travelling at close the terminal velocity, equal to 3d
because the cavity carved in the water by the impacting drop had a maximum depth equal to 3d (Engel,
1966). For flow depths less than about 4 mm ($1/h = 0.25$), the depth of flow imposes an appreciably
constraint on particle travel distance so that sediment concentrations divided by $1/h - 0.1$ are less than
predicted using the relationship for deeper flows .

Figure 9 shows the relationships between the sediment concentrations per unit rainfall intensity
for the soil monoliths and $1/h - 0.1$. The relative rankings for the susceptibility of the surfaces to
erosion using $1/h - 0.1$ and $0.0104 \, h^2 - 0.2017 \, h + 1.0$ are similar (Table 3). Equation 9 was also used to
determine the erodibility of microphyte-dominated calcareous soils in woodland near Wentworth, New
South Wales, Australia (Eldridge and Kinnell, 1997).

## 5 Discussion


The method described above is unique to the extent that no other method reported in the
literature provides as high a degree of control on the factors known to influence the erosive stress
applied to the soil surface when raindrops impact shallow surface water flows. In using the method, the
ranking of soils in respect to their susceptibility to erosion is obtained through an empirical factor that
results from experiments when flow depth is varied in a manner that controls the erosive stress applied
to the soil surface that is not achieved by laboratory and field experiments where natural or artificial
rain is applied to inclined surfaces. It can be argued that flow depth is, for a given rainfall, a soil specific
property (determined by infiltration, crust formation etc.), so that controlling runoff depth is not





required to estimate an erodibility coefficient commonly used in erosion models. However, as illustrated
by the erodibility data for WEPP shown in Table 2, that argument is hardly compelling.

The apparatus shown in Figure 2 can be used under artificial rainfall produced by sprays

provided that the depth effect is determined for that rainfall producing system. Spray rainfall simulators
are widely used because that have a wide drop size distribution that may be similar to those found in
rainfall but many different types of nozzle have been used without good knowledge of the variation of
the intensity and kinetic energy of the rain within the target area. As demonstrated by Iserloh et al
(2013) and Lassuet al ( 2014), the drop size, drop velocity and drop impact frequency characteristics of
rain produced by many rainfall simulators using nozzles are far from spatially uniform. Also, spatial
variations in rainfall characteristics can vary with the pressure of the water supplied to the nozzle and,
in many cases, the control of that pressure in not well maintained. As noted earlier, spatial variations in
rainfall characteristics particularly in the zone near the bottom end of the eroding surface can have an
appreciable influence of sediment discharge under rain-impacted flows. The spatial uniformity of the
rainfall characteristics, and control of raindrop size and energy, is better maintained in systems the
produce rain from drop formers such as described above.

Although the method is based on the model described by Eq. 2, the results from the experiments

reported above have been used in a qualitative assessment of the susceptibility of the eroding surfaces
rather than as quantitative values of erodibility that can be applied to predicting sheet and interrill
erosion even though equations that model the effect of flow depth on f[h,d] over the range of drop sizes
commonly observed in natural rainfall exists (Kinnell, 1993b). For a surface of uniform sized sand, the
particle size distribution of the eroding surface remains constant with time so that $k_{sd}$ remains constant
with time. However, as noted above, some soil particles detached by raindrops impacted flows travel in
the flow in suspension while others saltate or roll along the surface and as a consequence, particles of
different size travel at different rates from the point where there were initially mobilized. This, in effect,
results in fast moving particles being winnowed from the eroding surface so that the particle size
composition of the particles sitting on the surface changes with time. Also, loose particles sitting on the



surface provide a degree of protection against detachment of soil particles from cohesive soil surfaces
(Kinnell 2005b, 2006) with the result that $k_{sd}$ varies with time. As noted above, for each soil surface, 10
mins of rain was applied to at each of 4 flow depths with $I_d$ = 64 mm/h and u = 20 mm/s with flow
depths decreasing over the range 7.5 mm to 3.2 mm. Using the reverse sequence with flow depth
increasing over the range 3.2 to 7.5 mm resulted in a different value of $c_{sd}/I_d$ being obtained being
produced for a given flow depth (Kinnell et al, 1996) as a result of the temporal changes that occur on
the eroding soil surface. Varying the time of exposure to rain and the length of the eroding surface may
also result in different values of $c_{sd}/I_d$ being produced for a given flow. Erosion by rain-impacted flows
involves complex interactions between raindrops, flowing water and the soil surface so that the
susceptibility of the soil surface to erosion by rain-impacted flow varies in time and space even when
rain and flow characteristics do not.

Although the method described above can be used to provide a qualitative rather than

quantitative assessment of the susceptibility of the eroding surfaces to erosion by rain- impacted flows,
the high degree of control of the factors that affect the erosive stress provides an environment with a
potential to be used in the study of how factors like cohesion affect soil erodibility. It also has the
potential to be used in studies on how surfaces eroding by rain-impacted flows mobilize carbon and
chemical pollutants to flows that transport them across the landscape.

**6 Concluding remarks**

Given flow depth, flow velocity, raindrop size, raindrop velocity and raindrop impact frequency

influence the erosive nature of rain-impacted flows, it is necessary to control and measure these factors
when determining the susceptibility of soil to erosion by rain-impacted flow. The apparatus shown in
Figure 2 is designed to produce controlled and measurable flow conditions over the eroding surface.
Such conditions are not achievable when applying artificial rainfall on field plots. While sprays from
nozzles may produce drop-size distributions comparable to those that occur in natural rainfall, uniform





spatial distributions of rainfall intensity, drop sizes and velocities are seldom achieved. Producing drops
from pendent drop formers can produce rainfall that is more spatially uniform provided care is taken to
ensure that spatial variations produced by the fact that adjacent droppers do not necessarily produce
drops at the same rate are dealt with appropriately.

In most practical situations, drops produced from pendent drop formers do not achieve terminal

velocity before impacting flows over erodible surfaces. Consequently, a calibration equation for the
effect of flow depth needs to be obtained that differs from that shown in Figure 4 in many practical
cases. Once that calibration equation has been developed, it can, as illustrated in Figure 7 and Table 3,
be used obtain qualitative values of the susceptibility of the soil surfaces to erosion by rain-impacted
flow.

In the experiments with soil monoliths reported above, the event duration was arbitrarily set at

10 minutes and using a different event duration may produce different results given that exposure to
erosion by rain-impacted flow may cause appreciable changes to occur in the soil surface, particularly if
the surface has been recently disturbed. In addition to the development of surface crusts which can
cause changes in erodibility to occur during a rainfall event, particles travelling by raindrop-induced
saltation and rolling may provide a degree of protection against detachment by raindrop impact (Kinnell
2005b, 2006) and influence how the particle size characteristics of sediment discharged by rain-
impacted flow varies over time (Kinnell, 2009). These factors need to be considered when interpreting
results for experiments involving rain-impacted flows. Also, monitoring both sediment discharge and
composition may facilitate studies on the effects of factors such as soil chemistry and aggregate stability
on erosion by rain-impacted flows.





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



Wischmeier,W.C., and Smith, D.D. 1965. Predicting rainfall-erosion losses from cropland east of the

Rocky Mountains. Agricultural  Handbook  No. 282. US Dept Agric., Washington, DC.

Wischmeier,W.C., and Smith, D.D. 1978. Predicting rainfall erosion losses – a guide to conservation

planning. Agricultural  Handbook  No. 537. US Dept Agric., Washington, DC

.




Table 1. Soil erodibilities in SI units associated with the USLE published by Risse et al (1993) and the values observed by Kinnell and Risse (1998) for the USLE and USLE-M models

| Rank | location | pub K | location | obs K | Location | $K_{UM}$ | location | pub k | obs K | $K_{UM}$ |
|---|---|---|---|---|---|---|---|---|---|---|
| | | Erodibility (t hr MJ⁻¹ mm⁻¹) | | | | | | erodibility relative to that at Arnot | | |
| 1 | Arnot | 0.0026 | Arnot | 0.0031 | Arnot | 0.0088 | Arnot | 1.0 | 1.0 | 1.0 |
| 2 | Tifton | 0.0066 | Tifton | 0.0067 | Tifton | 0.0185 | Tifton | 2.5 | 2.2 | 2.1 |
| 3 | Watkinsville | 0.0237 | Guthrie | 0.0119 | Guthrie | 0.0256 | Watkinsville | 9.0 | 8.5 | 6.2 |
| 4 | Madison | 0.0290 | Presque Isle | 0.0162 | Statesville | 0.0504 | Madison | 11.0 | 17.9 | 11.8 |
| 5 | Guthrie | 0.0290 | Castana | 0.0262 | Presque Isle | 0.0536 | Guthrie | 11.0 | 3.8 | 2.9 |
| 6 | Presque Isle | 0.0303 | Watkinsville | 0.0264 | Watkinsville | 0.0547 | Presque Isle | 11.5 | 5.2 | 6.1 |
| 7 | Marcellus | 0.0369 | Statesville | 0.0270 | McCredie | 0.0728 | Marcellus | 14.0 | 12.6 | 9.5 |
| 8 | Morris | 0.0369 | McCredie | 0.0327 | Castana | 0.0737 | Morris | 14.0 | 11.1 | 15.2 |
| 9 | Castana | 0.0435 | Morris | 0.0345 | Marcellus | 0.0836 | Castana | 16.5 | 8.5 | 8.4 |
| 10 | Statesville | 0.0474 | Marcellus | 0.0390 | Holly Springs | 0.0933 | Statesville | 18.0 | 8.7 | 5.7 |
| 11 | La Crosse | 0.0500 | La Crosse | 0.0521 | La Crosse | 0.0996 | La Crosse | 19.0 | 16.8 | 11.3 |
| 12 | Holly Springs | 0.0672 | Madison | 0.0554 | Madison | 0.1037 | Holly Springs | 25.5 | 21.5 | 10.6 |
| 13 | McCredie | 0.0817 | Bathnay | 0.0619 | Bathnay | 0.1228 | McCredie | 31.0 | 11.1 | 8.3 |
| 14 | Bathnay | 0.1027 | Holly Springs | 0.0667 | Morris | 0.1337 | Bathnay | 39.0 | 20.0 | 14.0 |



Table 2 WEPP Interrill erodibilities for flat and ridged plots determined by Kinnell (1993b) from the data presented by Elliot et al (1989).

| | Interrill erodibility ($10^{-6}$ kg s m$^{-4}$) | | | | $K_i$ relative to Heiden | |
|---|---|---|---|---|---|---|
| rank | Location | Ki.ridge | Location | Ki.flat | Ki.ridge | Ki.flat |
| 1 | Heiden | 1.62 | Portneuf | 1.79 | 1.00 | 0.53 |
| 2 | Portneuf | 2.31 | Sverdrup | 2.00 | 1.43 | 0.59 |
| 3 | Sharpsberg | 2.51 | Barnes-ND | 2.02 | 1.55 | 0.60 |
| 4 | Barnes-MN | 2.58 | Pierre | 2.72 | 1.59 | 0.81 |
| 5 | Pierre | 2.72 | Barnes-MN | 2.84 | 1.68 | 0.84 |
| 6 | Barnes-ND | 3.06 | Woodward | 3.04 | 1.89 | 0.90 |
| 7 | Los Banos | 3.27 | Whitney | 3.10 | 2.02 | 0.92 |
| 8 | Williams | 3.55 | Academy | 3.32 | 2.19 | 0.99 |
| 9 | Zahl | 3.84 | Heiden | 3.37 | 2.37 | 1.00 |
| 10 | Academy | 3.97 | Keith | 3.82 | 2.45 | 1.13 |
| 11 | Sverdrup | 3.97 | Hersh | 3.89 | 2.45 | 1.15 |
| 12 | Keith | 4.26 | Los Banos | 4.13 | 2.63 | 1.23 |
| 13 | Whitney | 4.38 | Zahl | 4.19 | 2.70 | 1.24 |
| 14 | Nansene | 4.93 | Sharpsberg | 4.53 | 3.04 | 1.34 |
| 15 | Palouse | 5.20 | Nansene | 4.64 | 3.21 | 1.38 |
| 16 | Amarillo | 6.19 | Williams | 5.10 | 3.82 | 1.51 |
| 17 | Hersh | 6.80 | Amarillo | 5.51 | 4.20 | 1.64 |
| 18 | Woodward | 7.56 | Palouse | 6.41 | 4.67 | 1.90 |

Table 3. Relative susceptibility to erosion by 2.7 mm raindrops travelling at near terminal velocity impacting flows over soil monoliths in the apparatus shown in Figure 1

| | relative susceptability | |
|---|---|---|
| Soil | Eq. (10) | Eq.(11) |
| Ginninderra | 1.000 | 1.000 |
| Oakvale D | 0.719 | 0.700 |
| Oakvale IC | 0.345 | 0.351 |
| Oakvale OC | 0.077 | 0.081 |



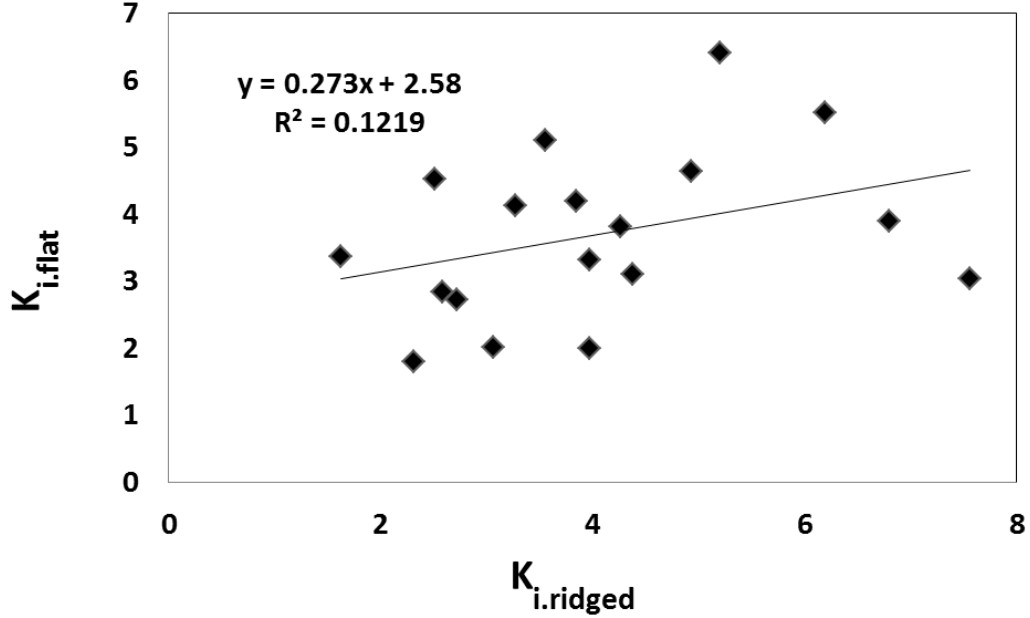

**Figure 1**. Relationship between flat and ridged erodibilities obtained by Kinnell (1993a) from the experiments of Elliot et al (1989).



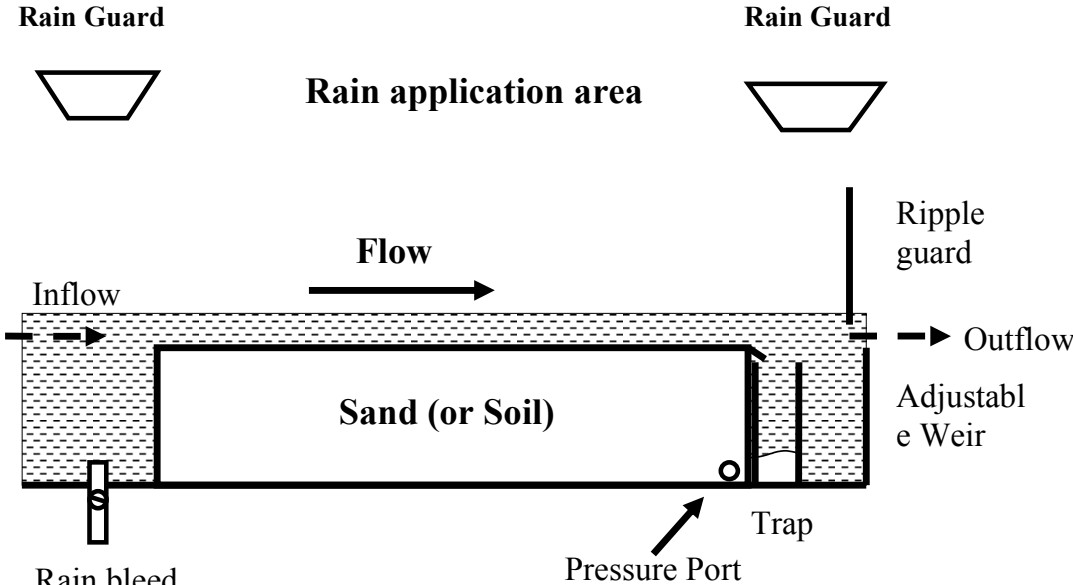

**Figure 2**. Apparatus used to determine the flow depth function for flows impacted by drops travelling at or near terminal velocity by Kinnell (1993b).



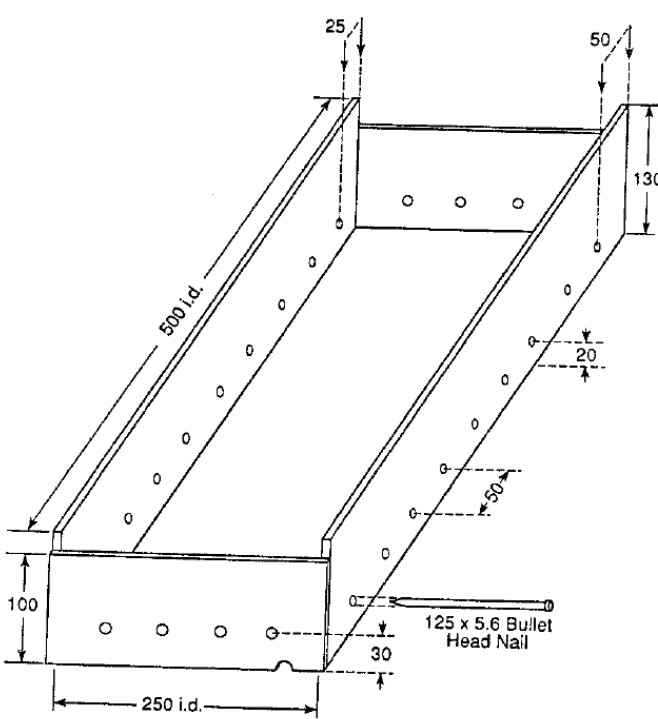

**Figure 3**. Sampling frame for collecting soil monoliths




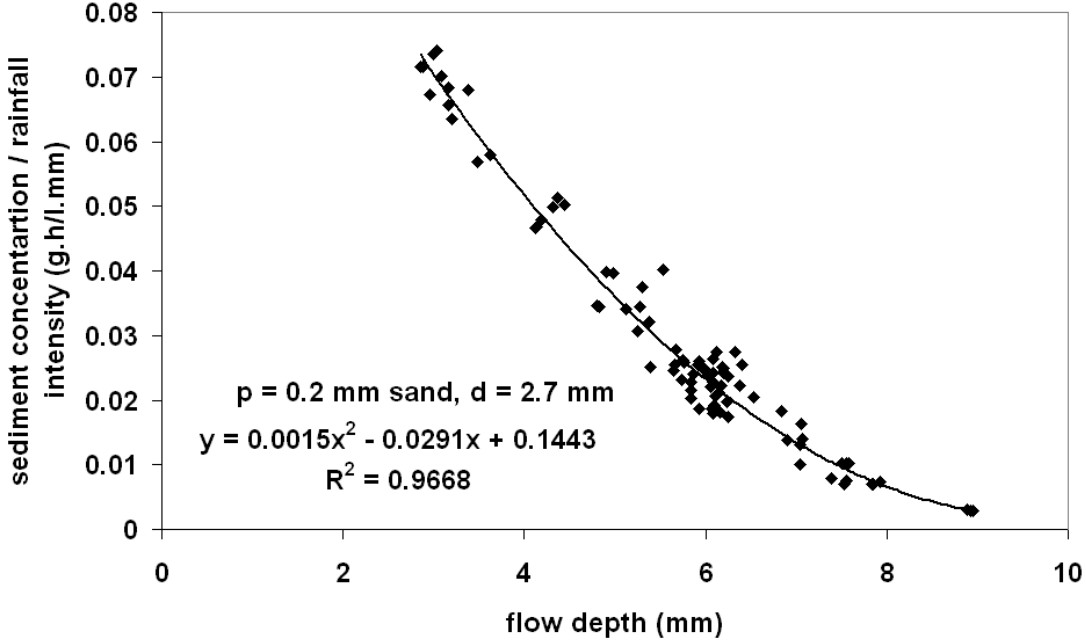

**Figure 4**. Relationship between sediment concentration per unit rainfall intensity and flow depth for 0.2 mm sand under flows impacted by 2.7 mm drops travelling at close to terminal velocity in experiments reported by Kinnell (1991, 1993b). (From Kinnell, 2009). Note the equation is not valid when h>9 mm.





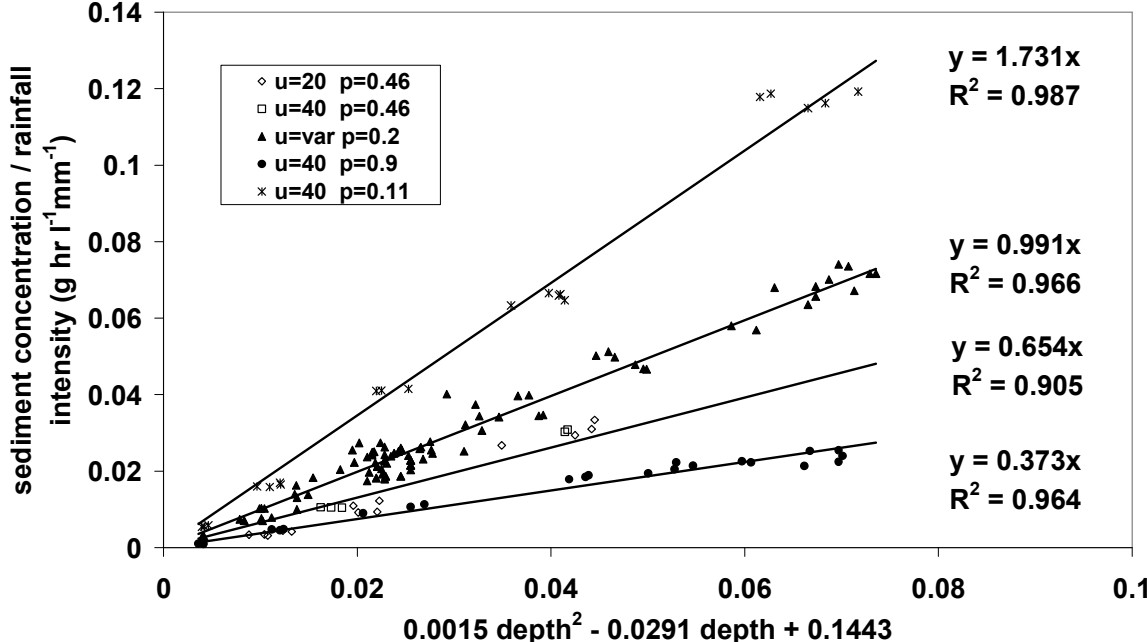

**Figure 5**. Relationships produced by Eq. (7) for beds of sand of various sizes eroding under flows impacted by 2.7 mm drops travelling at close to terminal velocity (from Kinnell, 2009).





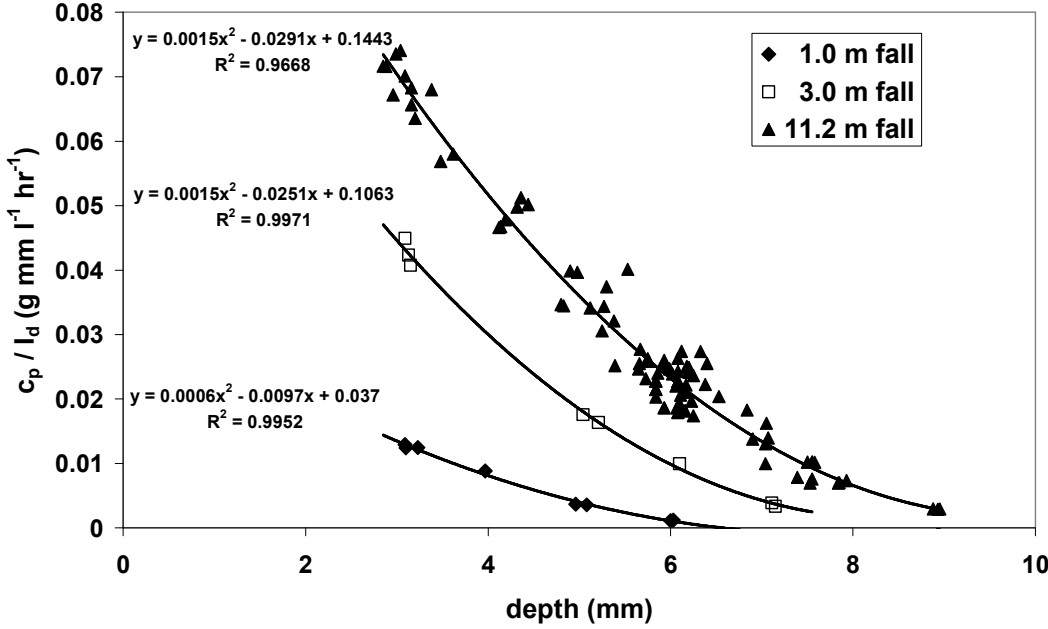

**Figure 6.** The effect of flow depth on sediment concentration per unit rainfall intensity for 0.2 mm sand under flows impacted by 2.7 mm drops falling from 1, 3 and 11.2 m (Kinnell, 2005a)





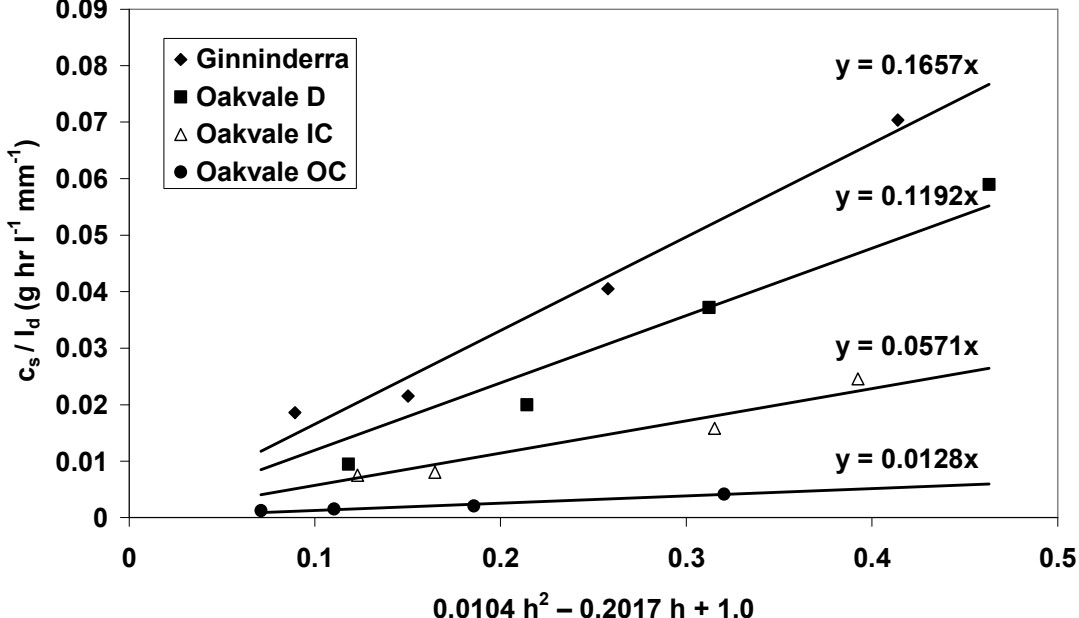

**Figure 7**. Relationships between 0.0104 h2 – 0.2017 h + 1.0 and sediment concentration per unit rainfall intensity for flows impacted by 2.7 mm drops travelling at near terminal velocity over soil monoliths (Kinnell, 2005b).




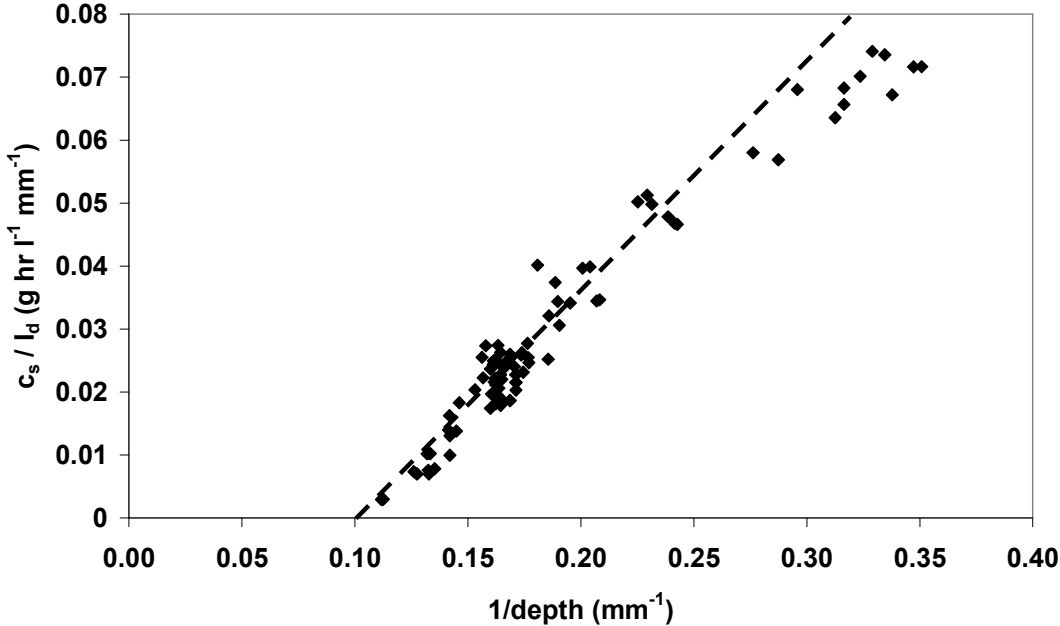

**Figure 8**. The relationship between sediment concentration per unit rainfall intensity and the inverse of flow depth for 2.7 mm drops travelling at close to terminal velocity impacting flows over 0.2 mm sand. The data are from the same experiments as used for Figure 4.





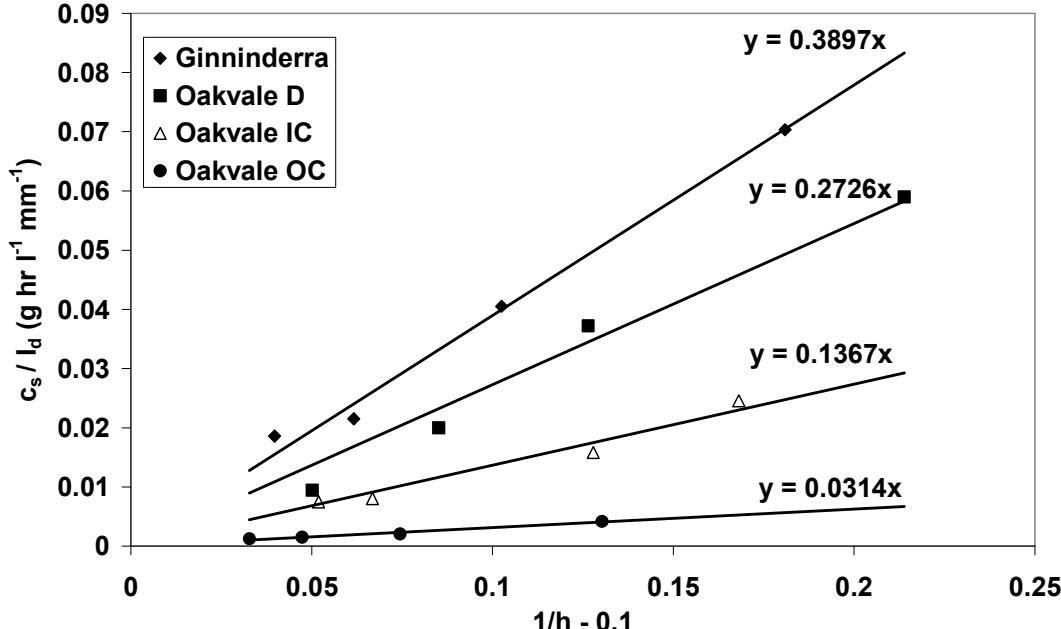

**Figure 9.** The relationship between sediment concentration per unit rainfall intensity and the inverse of flow depth minus 0.1 for 2.7 mm drops travelling at close to terminal velocity impacting flows over soil surfaces in the apparatus shown in Figure 1. The data are for the same experiments as used in Figure 7.