# Peer review of "Determining the susceptibility of soils materials to erosion by rain- 2 impacted flows"

_SOIL, 2016_

## Referee Comment (RC1) · Anonymous Referee #1 · 29 Feb 2016

The author re-visited some old data sets and used a simple framework to interpret the controlled experimental results. The author argues that the soil erodibility, or soil susceptibility to erosion, is very much a function of the theoretical and modeling framework in which the notion of erodibility or susceptibility is introduced and formulated. The author demonstrates how the susceptibility of soils could be determined in situations where the impact of rain drops is moderated by the depth of overland flow.

Here are my major concerns with the manuscript:

1) In the introduction section, much is written about various values of soil erodibility, and how these values depended on the model adopted, and the erodibility value for the same soil may be quite different in relative terms. It is worth sharing the insight that while soil erodibility is a useful concept, its value is certainly a function of the

theoretical and modeling framework, soil property, and the state that the soil is in. There is, however, very little in the introduction about what we know and we do not know about the moderating effect of water depth on the rainfall impact. A review of literature on rain-impacted flows is necessary to contextualise this short communication.

2) Comparing Eq.(1) and Eq.(2), the author basically assumes that the overland flow by itself is not able to erode, erosion (detachment) occurs entirely as a result of rainfall impact on ponded surfaces. Thus slope and flow rate, in the form of shear stress or stream power, do not feature in subsequent analysis. In many parts of world with serious soil erosion, the landscape is commonly rather steep and rugged. What is the impact of rainfall on soil detachment and on sediment-laden flows on steep slopes?

3) What are the objectives of the manuscript? Authors are required to make a clear statement in the introduction about the intent of the research and the objectives.

4) Without checking the author's publications in the early 1990s, and in the 2000s, it is fairly evident that most of the data and experiment results used for tables and diagrams have been published previously, especially Kinnell (1991, 1993a, 1993b, 2005a, 2005b, 2009). While there is nothing wrong to re-use and re-interpret previous experimental results, I would recommend that the author make absolutely clear what is the contribution that is new and original in the manuscript.

5) I guess that Table 3 was used to show the consistency in relative susceptibility. However, I could not find Eq. (10) and Eq. (11) in the text. Anyway, for controlled experiments, it won't be hard to have soil erodibilities in the right ranking order using two different modelling frameworks.

Apart from the issues outlined above, the paper is well written, with few typos and errors of omission.

---

## Author Comment (AC1) · 15 Mar 2016

The objective of the manuscript is to describe a method that could be used to determine the susceptibility of erosion to rain-impacted flows without including an in depth review of the mechanisms involved in erosion by rain-impacted flows (comments #1 and #3). Arguably, the title should begin with "A method for . . . . . . ." to make this objective more clear at the outset.

The reviewer comments on the fact that the method based on Eq. 2 is directed to the case where flow shear stress is not involved in causing detachment. Erosion is a two staged process, stage 1 being detachment, the plucking of soil material from within the soil surface where it is held by cohesion and interparticle friction, and stage 2, the transport of the detached material from the site of detachment. Certainly, there are

situations where detachment results from both raindrop impact and flow shear in the same area at the same time (eg within a shallow rill during rain), but generally, the effect of flow shear on detachment is not considered significant in modelling interrill and sheet erosion. Eq. 1 is based on the consideration that sediment discharge from interrill and sheer erosion areas is dependent on runoff and sediment concentration where sediment concentration is empirically dependent on rainfall intensity and slope gradient. Eq. 2 is again based on the product runoff and sediment concentration with sediment concentration being empirically related to a function which accounts for the interaction of flow depth and drop size which is also dependent on drop velocity. As noted above, the manuscript was intended to focus on experimental methodology rather than the physics leading to Eq. 2 which was described in, for example, Kinnell 1993b.

Certainly, the method has been described to some extent in a number of papers but this manuscript brings together relevant information that is scattered among the papers the reviewer listed so that a user or potential user of the method has that information in a single place. The application of the method to determining soil erodibility is not discussed in any depth in the existing literature and so the tables are new. However, as noted above, the objective is to present a method which is unique in that it is the only one described in the literature which controls the flow and rain characteristics that influence the erosive factors in rain-impacted flow well. Those controls not only important in the determination of the susceptibility of soil to erosion by rain-impacted flow but are also important in determining how soil surfaces eroding under rain-impacted flow act as sources of pollutants that affect water quality.

---

## Referee Comment (RC2) · Anonymous Referee #2 · 18 Mar 2016

Review SOIL-2016-05: Determining the susceptibility of soil materials to erosion by rain-impacted flows

*General comments:*

1. As already mentioned by referee 1 the objectives of the manuscript need to be formulated. It should also be made clear what is new in this manuscript compared to the older manuscripts of the author, so that one can understand why the manuscript is "worth" to be published as a "Short communication".

2. It should be made clearer for which purposes the method might be used in the future including references to some recent literature, where this method might be helpful. In lines 309-310 the author states that the method "…provides a qualitative rather than quantitative assessment of the susceptibility of the eroding surfaces…". To me it is not clear how this qualitative method (ranking of soil erodibility) can help in calculations of carbon mobilization or chemical pollutants (Line 313-314) mentioned by the author.

3. In general, the manuscript is clearly written. However, in several passages one single sentence extends over more than three lines. Please separate these sentences into several ones to improve understanding, e.g. lines 37-41, 286-290, 336-339, 339-343 and possibly more passages.

4. The derivation of equation 5 is not clear, in line 188 I guess equation 2 is meant instead of equation 1?

5. Line 235ff: I recommend making an overview table that shows the important parameters of the experiments and their variations.

---

## Author Comment (AC2) · 20 Mar 2016

Responses to Anonymous Referee #2

As noted in my responses to Anonymous Referee #1, the objective of the manuscript is to describe a method that can be used to determine the susceptibility of erosion to rain-impacted flows without including an in depth review of the mechanisms involved in erosion by rain-impacted flows. Arguably, the title should begin with "A method for ……." to make this objective more clear at the outset. The paper is in a sense a Technical Note (a category that does not exist in many journals these days) and, as such, the material presented is directed at describing the method and illustrating the product of the method. The need to know what is new and not new is not necessarily important to a reader who may be interested in using the method.

Eq (5) does, as noted by the reviewer, follow from Eq(2) [not Eq (1)] and the fact that water discharge is given by the product of flow depth (h) and flow velocity (h) and the paper was written based on the assumption that the reader knows that fact.

The comment made in the about the use of the method having a "potential to be used in studies on how surfaces eroding by rain-impacted flows mobilize carbon and chemical pollutants to flows that transport them across the landscape" was not intended to be associated with the qualitative ranking of the susceptibility soil surfaces. The comment was meant to suggest that "the high degree of control of the factors that affect the erosive stress means that the equipment may also have the potential to be used in studies on how surfaces eroding by rain-impacted flows mobilize carbon and chemical pollutants to flows that transport them across the landscape".
* * *